# Phenylpropanoid Content of Chickpea Seed Coats in Relation to Seed Dormancy

**DOI:** 10.3390/plants12142687

**Published:** 2023-07-19

**Authors:** Veronika Sedláková, Sanja Ćavar Zeljković, Nikola Štefelová, Petr Smýkal, Pavel Hanáček

**Affiliations:** 1Department of Plant Biology, Mendel University in Brno, 613 00 Brno, Czech Republic; veronika.sedlakova@mendelu.cz; 2Department of Genetic Resources for Vegetables, Medicinal and Special Plants, Crop Research Institute, 783 71 Olomouc, Czech Republic; zeljkovic@vurv.cz; 3Czech Advanced Technology and Research Institute, Palacký University, 783 71 Olomouc, Czech Republic; nikola.stefelova@upol.cz; 4Department of Botany, Faculty of Science, Palacký University, 783 71 Olomouc, Czech Republic

**Keywords:** chickpea, dormancy, seed coat, legumes, phenolic acids, flavonoids

## Abstract

The physical dormancy of seeds is likely to be mediated by the chemical composition and the thickness of the seed coat. Here, we investigate the link between the content of phenylpropanoids (i.e., phenolics and flavonoids) present in the chickpea seed coat and dormancy. The relationship between selected phenolic and flavonoid metabolites of chickpea seed coats and dormancy level was assessed using wild and cultivated chickpea parental genotypes and a derived population of recombinant inbred lines (RILs). The selected phenolic and flavonoid metabolites were analyzed via the LC-MS/MS method. Significant differences in the concentration of certain phenolic acids were found among cultivated (*Cicer arietinum*, ICC4958) and wild chickpea (*Cicer reticulatum*, PI489777) parental genotypes. These differences were observed in the contents of gallic, caffeic, vanillic, syringic, *p*-coumaric, salicylic, and sinapic acids, as well as salicylic acid-2-O-β-d-glucoside and coniferaldehyde. Additionally, significant differences were observed in the flavonoids myricetin, quercetin, luteolin, naringenin, kaempferol, isoorientin, orientin, and isovitexin. When comparing non-dormant and dormant RILs, significant differences were observed in gallic, 3-hydroxybenzoic, syringic, and sinapic acids, as well as the flavonoids quercitrin, quercetin, naringenin, kaempferol, and morin. Phenolic acids were generally more highly concentrated in the wild parental genotype and dormant RILs. We compared the phenylpropanoid content of chickpea seed coats with related legumes, such as pea, lentil, and faba bean. This information could be useful in chickpea breeding programs to reduce dormancy.

## 1. Introduction

Chickpea (*Cicer arietinum* L.) is an annual grain legume crop adapted to dry climates and is the second most important pulse in terms of its production and consumption by humans. Cultivated chickpeas are divided into two types—*desi* and *kabuli*. The *desi* type has small, irregularly shaped seeds with thicker and colored seed coats, in which the seed coat accounts for approximately 14% of the total weight of the seed. The *kabuli* type produces comparatively larger and smoother round seeds with thinner unpigmented seed coats; the seed coat forms approximately 5% of the total weight of the seed [1,2]. The larger-seeded *kabuli* type was likely derived from the smaller-seeded *desi* type [3,4], while seeds of the *desi* type are very similar to the seeds of the ancestor of cultivated chickpea, *C. reticulatum* Ladiz. [5]. As in other pulses, chickpea seeds are an excellent source of high-quality proteins and also contain a variety of phytochemicals and other bioactive compounds [6,7]. Seeds of legumes also contain phenylpropanoids, specifically phenolic acids and flavonoids [8,9,10,11]. Most of these compounds are concentrated in the seed coats [8,12,13] and have important roles in many physiological and metabolic processes [8]. A phytochemical profiling study of legumes by Tor-Roca et al. [7] identified a total of 478 phytochemicals, of which 405 (85%) were phenylpropanoids (13% phenolic acids and 40% flavonoids). The contents of many of these metabolites were reduced during domestication [14], whereas in the wild, phenylpropanoids play an important role in the defense of the plant against various pathogens.

Physical or seed-coat-imposed dormancy, which occurs in wild legume ancestors, is caused by an impermeable barrier in the seed coat formed by specific, yet-to-be-identified compounds [15]. Both the composition and structure of the seed coat have been reported to influence dormancy [16,17,18]. Importantly, pigmentation of the seed coat correlates, to a certain extent, with water uptake during germination, with pigmented seeds imbibing slower and germinating later than non-pigmented seeds [19,20,21,22]. This has also been shown in legumes, including peas [20,23], common beans [24,25], faba beans [26], soybeans [27], and chickpeas [28]. On the other hand, genetic analysis using wild, pigmented, and cultivated non-pigmented pea genotypes [17] has shown that seed coat pigmentation and dormancy can be decoupled and are only associated when comparing contrasting wild and cultivated parents.

Among the plant metabolites, phenylpropanoids have been shown to affect seed longevity and dormancy by making the seed coat impermeable [15,18,20,23,29]. Higher contents of phenolic compounds and an increased activity of oxidizing enzymes has been found in wild pea seeds compared with non-dormant cultivated pea seeds [11,17,18,20,23].

Generally, phenylpropanoids are a large and very versatile class of secondary metabolites that are synthesized through the shikimate/phenylpropanoid biosynthetic pathway. Their biosynthesis starts with two amino acids, phenylalanine and tyrosine, which produce 4-coumaroyl-CoA, a central intermediate in the biosynthesis of phenylpropanoids [30]. Hydroxycinnamates are the first products of the phenylpropanoid pathway and represent the simplest natural phenolics, with a carbon skeleton of C6–C3. Next are hydroxybenzoic acids with a C6–C1 skeleton, which are primarily formed through the reduction of hydroxycinnamic acids [31], although the enzymes involved in these transformations have still not been characterized. Importantly, 4-coumaroyl-CoA, together with malonyl-CoA creates chalcones, which are biosynthetic precursors of flavonoids, the largest class of phenylpropanoids. Based on their biosynthesis and structures, flavonoids can be divided into six groups: flavones, flavonols, flavan-3-ols, flavanonols, flavanones, isoflavones, and anthocyanins. All of these groups contain the core flavan structure [32]. Figure 1 shows a representation of the simplified phenylpropanoid biosynthesis, where only the compounds detected in this study are presented.

Phenylpropanoids play highly important roles in plant development (such as organ pigmentation, pollen fertility, and symbiotic nitrogen fixation), but also in plant environment biotic and abiotic interactions (e.g., pollinator attraction, pathogen, and herbivore resistance, reduction of reactive oxygen, and UV protection) [33]. Several studies have reported the inhibitory effect of phenolic substances on the germination of seeds. In the case of *Sapium sebiferum* (L.) Roxb. (*Euphorbiaceae* Juss.), extracts of the seed coat containing proanthocyanidins significantly inhibited germination via altered gene expression [34]. The same was observed in *Arabidopsis* Heynh. transparent testa mutants [35], yellow-seeded rapeseed (*Brassica napus* L.), and flax mutants (*Linum usitatissimum* L.) [36,37]. Comparative analysis of wild and cultivated soybean seed coats identified epicatechin as influencing seed dormancy [15]; this was recently supported by the larger analysis of Li et al. [38]. Similarly, genes governing flavonoid biosynthesis pathways in seed pigmentation were found to affect dormancy in *Arabidopsis* seeds. This occurred through the regulation of the expression of anthocyanidin reductase, the enzyme responsible for the epicatechin building block of proanthocyanidins [16,39,40].

In our previous study, we showed that dormancy is not directly related to the thickness of the seed coat in chickpea [4], and it is more likely to be mediated by the chemical composition of the seed coat. Here, we aim to take our investigation of dormancy a step further by analyzing the relative concentration of selected phenolic acids and flavonoids in the seed coats of wild and cultivated genotypes and derived recombinant inbred lines of the chickpea. 

## 2. Results

The chickpea dormancy status was determined by Sedláková et al. [4], where the imbibition and germination were tested, and several gemination coefficients were calculated. The germination percentage of cultivated ICC4958 was 100% after 24 h, while for wild PI489777 it was 36.1% at 24 h, 60.2% at 72 h, 90.9% at 10 days, and 100% at 30 days. Recombinant inbred lines showed a broader distribution. We have selected the group of 16 non-dormant recombinant inbred lines (RILs) for which germination percentages ranged from 70% to 100% at 24 h, 78.5% to 100% at 72 h, 88% to 100% at 10 days, and 89% to 100% at 30 days. We have also selected the contrasting group of 12 dormant RILs for which the germination percentage ranged from 2% to 46% at 24 h, 4% to 56.9% at 72 h, 22% to 83.9% at 10 days, and 24% to 100% at 30 days.

### 2.1. The Content of Phenolic Compounds and Their Derivatives

In total, 24 phenolic compounds and their derivatives were analyzed. Of these, 13 were present in the seed coats of both parental genotypes, except for vanillic acid, which was not detected in cultivated ICC4958, while its level in wild PI489777 was 0.929 ± 0.136 pmol/mg. The average contents and standard deviations of the measured phenolic compounds are shown in Table 1; maximal values are stated in the text. The most abundant phenolic acid in both parental genotypes was 4-hydroxybenzoic acid. In cultivated ICC4958, this phenolic acid reached a concentration of up to 8.963 pmol/mg (average 8.303 ± 0.865 pmol/mg). The second most abundant was gallic acid: up to 1.862 pmol/mg (1.715 ± 0.157 pmol/mg). Values an order of magnitude lower were found for ferulic acid (up to 0.705 pmol/mg, 0.678 ± 0.026 pmol/mg), salicylic acid (up to 0.620 pmol/mg, 0.580 ± 0.006 pmol/mg), salicylic acid-2-O-β-d-glucoside (up to 0.531 pmol/mg, 0.508 ± 0.033 pmol/mg), chlorogenic acid (up to 0.468 pmol/mg, 0.438 ± 0.033 pmol/mg), *p*-coumaric acid (up to 0.391 pmol/mg, 0.389 ± 0.002 pmol/mg), and sinapic acid (up to 0.118 pmol/mg, 0.114 ± 0.004 pmol/mg). The least abundant phenolic compounds in cultivated ICC4958 were 3-hydroxybenzoic acid, caffeic acid, syringic acid, and coniferaldehyde. In the case of wild PI489777, the most abundant phenolic acid was 4-hydroxybenzoic acid, with a concentration of up to 11.036 pmol/mg (average 9.645 ± 1.350 pmol/mg). High levels were measured for salicylic acid-2-O-β-d-glucoside (up to 5.002 pmol/mg, 4.258 ± 0.809 pmol/mg), gallic acid (up to 2.501 pmol/mg, 2.336 ± 0.156 pmol/mg), salicylic acid (up to 2.120 pmol/mg, 1.870 ± 0.226 pmol/mg), and vanillic acid (up to 1.080 pmol/mg, 0.929 ± 0.136 pmol/mg). Values an order of magnitude lower were measured with ferulic acid (up to 0.798 pmol/mg, 0.692 ± 0.119 pmol/mg), chlorogenic acid (up to 0.582 pmol/mg, 0.504 ± 0.100 pmol/mg), *p*-coumaric acid (up to 0.215 pmol/mg, 0.187 ± 0.029 pmol/mg), and sinapic acid (up to 0.175 pmol/mg, 0.159 ± 0.019 pmol/mg). The least abundant phenolic compounds detected in wild PI489777 were 3-hydroxybenzoic acid, syringic acid, caffeic acid, and coniferaldehyde.

A significant difference in levels was found between parental genotypes for gallic acid, salicylic acid-2-O-β-d-glucoside, vanillic acid, syringic acid, salicylic acid, caffeic acid, *p*-coumaric acid, sinapic acid, and coniferaldehyde (Figure 2). Higher levels of gallic acid (2.336 ± 0.156 pmol/mg), salicylic acid (1.870 ± 0.226 pmol/mg), salicylic acid-2-O-β-d-glucoside (4.258 ± 0.809 pmol/mg), vanillic acid (0.929 ± 0.136 pmol/mg), syringic acid (0.043 ± 0.011 pmol/mg), sinapic acid (0.159 ± 0.019 pmol/mg), and coniferaldehyde (0.045 ± 0.004 pmol/mg) were detected in the wild PI489777. In contrast, higher levels of caffeic acid (0.054 ± 0.002 pmol/mg) and *p*-coumaric acid (0.389 ± 0.002 pmol/mg) were found in the cultivated ICC4958. Differences in other measured phenolic acids were not statistically significant; nevertheless, the levels of all phenolic acids with no significance were higher in PI489777. Apart from vanillic acid, the greatest difference between parental genotypes was observed in the content of salicylic acid-2-O-β-d-glucoside.

The levels of phenolic compounds in the seed coats of non-dormant RILs are summarized in Appendix A. Dormant RILs plus intermediate CRIL2-25 are summarized in Appendix A. Vanillic acid was not detected in the seed coats of six non-dormant RILs, similarly to its absence in cultivated non-dormant ICC4958. On the other hand, vanillic acid was also not detected in two dormant RILs and the intermediate (in terms of dormancy status) CRIL2-25 line. This CRIL2-25 line was subsequently excluded from further statistical analyses. 3-Hydroxybenzoic acid was not detected in the seed coat of one non-dormant RIL. In contrast, gallic acid was not detected in the seed coats of eight dormant RILs, which is more than half of the dormant RILs, but it was present in both parental genotypes with the dormant parent PI489777 exhibiting a higher level. The contents of phenolic acids such as chlorogenic, caffeic, vanillic, 3-hydroxybenzoic, syringic, *p*-coumaric, and ferulic acids were rather constant in non-dormant lines. Gallic acid was highly abundant in non-dormant CRIL2-5 (14.240 ± 0.968 pmol/mg) and CRIL2-6 (10.705 ± 1.199 pmol/mg) in comparison with the minimum value (0.673 ± 0.032 pmol/mg) for the respective category. Non-dormant CRIL2-45 (6.013 ± 0.032 pmol/mg) and CRIL2-50 (8.148 ± 1.557 pmol/mg) showed significant levels of salicylic acid-2-O-β-d-glucoside in comparison with the minimum value (0.110 ± 0.015 pmol/mg) of the respective category. Non-dormant CRIL2-7 (11.239 ± 1.762 pmol/mg), CRIL2-43 (10.130 ± 0.707 pmol/mg), and CRIL2-80 (10.778 ± 1.939 pmol/mg) were abundant in 4-hydroxybenzoic acid in comparison with the minimal value (2.163 ± 0.277 pmol/mg) of the respective category. Non-dormant CRIL2-43 (2.799 ± 0.280 pmol/mg), CRIL2-50 (2.566 ± 0.391 pmol/mg), and CRIL2-80 (2.960 ± 0.824 pmol/mg) were abundant in salicylic acid in comparison with minimal value (0.244 ± 0.051 pmol/mg) in the respective category. Non-dormant CRIL2-111 (1.551 ± 0.256 pmol/mg) was abundant in sinapic acid when compared with the minimal value (0.054 ± 0.012 pmol/mg) of the respective category (Appendix A). Dormant CRIL2-106 (5.787 ± 0.489 pmol/mg) and CRIL2-114 (5.056 ± 0.459 pmol/mg) were abundant in salicylic acid-2-O-β-d-glucoside in comparison with the minimal value (1.606 ± 0.186 pmol/mg) of the respective category. Dormant CRIL2-60, CRIL2-106, CRIL2-129, and CRIL25-131 (average 12.76 ± 0.133 pmol/mg) were highly abundant in 4-hydroxybenzoic acid in comparison with the minimal value (4.518 ± 0.600 pmol/mg) in the respective categories. Dormant CRIL2-114 (2.291 ± 0.231 pmol/mg) was abundant in salicylic acid in comparison with the minimal value (0.396 ± 0.064 pmol/mg) in the respective category (Appendix A).

Selected non-dormant RILs with 100% germination after 24 h (e.g., CRIL2-5, CRIL2-6, CRIL2-7, CRIL2-50, CRIL2-65, and CRIL2-89) showed differences between the levels of phenolics gallic acid (ranging from 1.190 to 14.240 pmol/mg), salicylic acid-2-O-β-d-glucoside (ranging from 0.110 to 8.148 pmol/mg), 4-hydroxybenzoic acid (ranging from 2.163 to 11.239 pmol/mg) and salicylic acid (ranging from 0.244 to 2.566 pmol/mg) (Appendix A). Chlorogenic, caffeic, vanillic, 3-hydroxybenzoic, syringic, *p*-coumaric, ferulic, and sinapic acids were constant. Selected dormant RILs with germination up to 20% after 24 h and up to 81.8% after 30 days (e.g., CRIL2-42, CRIL2-60, CRIL2-106, CRIL25-129, and CRIL2-131) showed fewer differences between phenolics than non-dormant RILs. The principal difference was the absence of gallic acid, except for CRIL2-106. CRIL2-42 differed in its level of 4-hydroxybenzoic acid, while values of the other selected dormant RILs were constant. Ferulic acid also differed slightly and ranged from 0.987 to 1.991 pmol/mg. Salicylic acid-2-O-β-d-glucoside, chlorogenic, caffeic, vanillic, 3-hydroxybenzoic, syringic, *p*-coumaric, and sinapic acids were constant (Appendix A).

The average content, standard deviation, and minimal and maximal values of measured data for the non-dormant RIL groups are shown in Table 2 and for the dormant RILs group in Table 3. The Wilcoxon test was applied to investigate the differences between the non-dormant and dormant RILs group. Significant differences between the levels of phenolic acids of seed coats of non-dormant and dormant RILs were found in gallic acid, 3-hydroxybenzoic acid, syringic acid, and sinapic acid (Wilcoxon test, *p* < 0.05). Higher levels of 3-hydroxybenzoic acid (0.146 ± 0.048 pmol/mg) were in dormant RILs, while higher levels of gallic acid (3.814 ± 3.814 pmol/mg), syringic acid (0.043 ± 0.018 pmol/mg), and sinapic acid (0.369 ± 0.357 pmol/mg) were in non-dormant RILs.

Boxplots of four phenolic compounds with significantly different levels (gallic acid, 3-hydroxybenzoic acid, syringic acid, and sinapic acid) between all non-dormant and dormant RILs are shown in Figure 3. Within the non-dormant and dormant RIL groups, outlying observations are marked with an asterisk.

There is variability among groups of non-dormant and dormant RILs in their germination times. The measured values of phenolic acids of selected RILs differed substantially from each other: three non-dormant lines (CRIL2-5, CRIL2-6, and CRIL2-50; 100% final germination percentage (FGP) after 24 h) and three dormant lines (CRIL2-106, CRIL2-60, and CRIL2-131; up to 50% FGP after 30 days) were selected and compared. These lines are considered to be the most representative of the given non-dormant and dormant category in terms of seed responses to water uptake and dormancy levels. The comparison was made for four phenolic acids (gallic acid, 3-hydroxybenzoic acid, syringic acid, and sinapic acid) and their levels showed the most significant changes in the non-dormant and dormant categories. Selected non-dormant and dormant RILs are marked in boxplots (Figure 3); their average contents are represented by black dots. The average concentrations of gallic acid, syringic acid, and sinapic acid were higher in three selected non-dormant RILs. On the other hand, 3-hydroxybenzoic acid had a higher average concentration in three selected dormant RILs.

#### Relationship of Non-Dormant and Dormant RILs to Parental Genotypes

The measured contents of phenolic acids in the group of non-dormant RILs corresponded with a parental genotype of the cultivated chickpea (ICC4958). For both cultivated ICC4985 and non-dormant RILs, the most abundant phenolics are 4-hydroxybenzoic and gallic acid with high concentrations of salicylic acid-2-O-β-d-glucoside and salicylic acid. The correspondence between the non-dormant RILs and the cultivated parental genotype was also demonstrated by syringic, caffeic, and 3-hydroxybenzoic acids being the least abundant phenolic compounds (Table 1 and Table 2).

The amount of phenolic acids in the group of dormant RILs corresponded to a parental genotype of the wild chickpea (PI489777). For both wild PI489777 and dormant RILs, the most abundant phenolic acid was 4-hydroxybenzoic acid, with very high concentrations of salicylic acid-2-O-β-d-glucoside, gallic, salicylic, and vanillic acids. On the other hand, caffeic, syringic, and 3-hydroxybenzoic acids were the least abundant phenolics in both wild PI489777 and dormant RILs (Table 1 and Table 3). The concentrations were similar in the cultivated vs. wild and non-dormant vs. dormant categories.

### 2.2. Contents of Flavonoids and Their Glycosides

In total, 26 flavonoids were analyzed. Of these, 13 were detected in the seed coats of both parental genotypes, except for catechin, which was not detected in wild PI489777, while its level in ICC4958 was 0.273 ± 0.041 pmol/mg. 

The average contents and standard deviations of the measured flavonoids are shown in Table 4; maximal values are stated further in the text. The most abundant flavonoids detected in cultivated ICC4958 were myricetin, with a concentration of up to 18.686 pmol/mg (average 17.668 ± 0.895 pmol/mg), and gallocatechin, with a concentration of up to 12.039 pmol/mg (11.196 ± 1.890 pmol/mg). Highly abundant were naringenin (up to 5.405 pmol/mg, 4.740 ± 0.984 pmol/mg) and myricitrin (up to 1.098 pmol/mg, 0.892 ± 0.294 pmol/mg). Values an order of magnitude lower were measured with quercitrin (up to 0.851 pmol/mg, 0.720 ± 0.114 pmol/mg), luteolin (up to 0.370 pmol/mg, 0.349 ± 0.033 pmol/mg), catechin (up to 0.320 pmol/mg, 0.273 ± 0.041 pmol/mg), and morin (up to 0.156 pmol/mg, 0.136 ± 0.025 pmol/mg). Very low abundance was measured with quercetin (up to 0.064 pmol/mg, 0.064 ± 0.001 pmol/mg), kaempferol (up to 0.047 pmol/mg, 0.040 ± 0.006 pmol/mg), and orientin (up to 0.030 pmol/mg, 0.028 ± 0.003 pmol/mg). The least abundant flavonoids detected in cultivated ICC4958 were isoorientin and isovitexin. In contrast, the most abundant flavonoids in wild PI489777 were gallocatechin, with a concentration of up to 10.568 pmol/mg (average 10.309 ± 0.224 pmol/mg), and myricetin, with a concentration of up to 9.344 pmol/mg (7.806 ± 1.619 pmol/mg). Naringenin was especially abundant (up to 2.460 pmol/mg, 2.310 ± 0.130 pmol/mg). Values 10 times lower were found for quercitrin (up to 0.979 pmol/mg, 0.827 ± 0.169 pmol/mg), myricitrin (up to 0.781 pmol/mg, 0.690 ± 0.097 pmol/mg), luteolin (up to 0.190 pmol/mg, 0.177 ± 0.013 pmol/mg), and morin (up to 0.152 pmol/mg, 0.115 ± 0.040 pmol/mg). The least abundant flavonoids measured in wild PI489777 were orientin, isoorientin, quercetin, isovitexin, and kaempferol. The most abundant flavonoids measured for both parental genotypes were gallocatechin and myricetin; the most significant difference was in the absence of catechin in wild PI489777.

Significant differences in the levels of flavonoid content in seed coats between parental genotypes were found for catechin, myricetin, quercetin, kaempferol, luteolin, isoorientin, orientin, isovitexin, and naringenin (Figure 4). Higher levels of isoorientin (0.050 ± 0.012 pmol/mg), orientin (0.085 ± 0.085 pmol/mg), and isovitexin (0.020 ± 0.020 pmol/mg) were measured in wild PI489777. In contrast, higher levels of myricetin (17.668 ± 0.895 pmol/mg), quercetin (0.064 ± 0.001 pmol/mg), luteolin (0.349 ± 0.033 pmol/mg), naringenin (4.740 ± 0.984 pmol/mg), and kaempferol (0.040 ± 0.006 pmol/mg) were identified in cultivated ICC4958. Differences in other measured flavonoids were not statistically significant. Apart from catechin, the greatest difference between parental genotypes was observed in the content of myricetin.

The levels of flavonoids in the seed coats of non-dormant RILs are summarized in Appendix A. Dormant RILs plus intermediate CRIL2-25 are summarized in Appendix A. The absence of catechin in dormant RILs could correspond with its absence in wild PI489777, since catechin was absent in 7 RILs of a total of 12, which is in more than half of dormant RILs. In the case of non-dormant RILs, 9 RILs of a total of 16 non-dormant RILs contained catechin, and fewer than half did not. In intermediate CRIL2-25, catechin was absent. Orientin was not detected in 2 non-dormant RILs and 1 dormant RIL. Isoorientin was not detected in 1 non-dormant RIL. Intermediate CRIL2-25 was excluded from subsequent statistical analyses. The contents of flavonoids such as catechin, myricitrin, quercitrin, quercetin, luteolin, naringenin, kaempferol, isoorientin, orientin, and isovitexin were rather constant in non-dormant RILs. Non-dormant CRIL2-6 was highly abundant in gallocatechin (17.630 ± 1.918 pmol/mg) and myricetin (20.010 ± 0.594 pmol/mg), in comparison with the minimal value (5.720 ± 0.576 pmol/mg) or (2.542 ± 0.481 pmol/mg) in the respective category. Non-dormant CRIL2-23 (13.068 ± 0.716 pmol/mg), CRIL2-43 (15.149 ± 2.875 pmol/mg) and CRIL2-51 (11.589 ± 1.102 pmol/mg) were abundant in morin in comparison with the minimal value (0.109 ± 0.008 pmol/mg) in the respective category (Appendix A). The contents of flavonoids of dormant RILs were rather constant in the cases of catechin, myricitrin, quercitrin, quercetin, naringenin, kaempferol, isoorientin, orientin, and isovitexin. Dormant CRIL2-48 and CRIL25-131 were highly abundant (16.731 ± 2.832 and 17.576 ± 0.225 pmol/mg) in gallocatechin in comparison with the minimal value (5.621 ± 1.072 pmol/mg) in the respective category. Dormant CRIL2-60 (11.566 ± 1.566 pmol/mg) and CRIL2-129 (15.145 ± 2.349 pmol/mg) were abundant in myricetin in comparison with the minimal value (2.251 ± 0.614 pmol/mg) in their respective categories. Dormant CRIL2-48 (11.932 ± 2.706 pmol/mg) was highly abundant in morin in comparison with the minimal value (0.114 ± 0.026 pmol/mg) in its respective category. Dormant CRIL2-48 (5.925 ± 1.507 pmol/mg) was abundant in luteolin in comparison with the minimal value (0.337 ± 0.091 pmol/mg) in its respective category (Appendix A).

Selected non-dormant RILs with 100% germination after 24 h (CRIL2-5, CRIL2-6, CRIL2-7, CRIL2-50, CRIL2-65, and CRIL2-89) showed differences in the levels of flavonoids gallocatechin (5.916–17.630 pmol/mg), quercitrin (0.632–3.174 pmol/mg), myricetin (2.542–12.550 pmol/mg), quercetin (0.029–0.637 pmol/mg), luteolin (0.176–1.925 pmol/mg), naringenin (2.676–8.754 pmol/mg), and morin (0.213–8.807 pmol/mg). On the other hand, myricitrin, kaempferol, isoorientin, orientin, and isovitexin did not vary significantly. Catechin was detected in half of the RILs; its levels were constant (Appendix A). Selected dormant RILs with germination up to 20% after 24 h and up to 81.8% after 30 days, CRIL2-42, CRIL2-60, CRIL2-106, CRIL25-129, and CRIL2-131, showed differences between the levels of flavonoids myricitrin (ranging from 1.153 to 3.049 pmol/mg), quercitrin (ranging from 0.900 to 2.958 pmol/mg), myricetin (ranging from 7.160 to 5.143 pmol/mg), luteolin (ranging from 0.337 to 1.511 pmol/mg), and morin (ranging from 0.210 to 3.329 pmol/mg). Gallocatechin, catechin, quercetin, naringenin, kaempferol, isoorientin, orientin, and isovitexin were constant (Appendix A).

The average contents, standard deviations, and minimal and maximal values of data for non-dormant RILs are shown in Table 5 and for dormant RILs in Table 6. The Wilcoxon test was applied to investigate differences between the non-dormant and dormant RILs group. Significant differences between seed coats of non-dormant and dormant RILs were found for quercitrin, quercetin, naringenin, kaempferol, and morin (Wilcoxon test, *p* < 0.05). Higher average levels of quercitrin (2.416 ± 1.186 pmol/mg), quercetin (0.321 ± 0.302 pmol/mg), naringenin (5.014 ± 2.265 pmol/mg), kaempferol (0.248 ± 0.243 pmol/mg), and morin (6.205 ± 4.974 pmol/mg) were measured in non-dormant RILs.

Boxplots of the five flavonoids (quercitrin, quercetin, naringenin, kaempferol, and morin) with significantly different levels compared between all non-dormant and dormant RILs are shown in Figure 5. Within the non-dormant and dormant RIL groups, outlying observations are marked with an asterisk.

There was variability among the groups of non-dormant and dormant RILs in their germination times. The measured values of flavonoids of selected RILs differed substantially from each other; therefore, three non-dormant RILs (CRIL2-5, CRIL2-6, and CRIL2-50; 100% FGP after 24 h) and three dormant RILs (CRIL2-106, CRIL2-60, and CRIL2-131; up to 50% FGP after 30 days) were selected and compared. These lines are considered to be the most representative lines of the given non-dormant and dormant category in terms of seed responses to water uptake and dormancy levels. The comparison was performed for five flavonoids (quercitrin, quercetin, naringenin, kaempferol, and morin); their levels showed the most significant changes in the non-dormant and dormant categories. Selected non-dormant and dormant RILs are marked in boxplots (Figure 5); their average contents are represented by black dots. All five flavonoids (quercitrin, quercetin, naringenin, kaempferol, and morin) had higher average concentrations in the three selected non-dormant RILs.

#### Relationship of Non-Dormant and Dormant RILs to Parental Genotypes

The measured contents of flavonoids in the group of non-dormant RILs corresponded to parental genotypes of the cultivated chickpea (ICC4958). For both cultivated ICC4958 and non-dormant RILs, the most abundant flavonoids were myricetin and gallocatechin, and the least abundant were isovitexin and isoorientin (Table 4 and Table 5).

Similarly, the amounts of flavonoids in the group of dormant RILs corresponded to the parental genotype of the wild chickpea (PI489777). For both wild PI489777 and dormant RILs, the most abundant flavonoids were gallocatechin and myricetin, with considerably higher concentrations in dormant RILs than in the wild PI489777. A very high concentration of naringenin was detected in both the wild PI489777 and in the dormant RILs. On the other hand, quercetin, isovitexin, isoorientin, and kaempferol are the least abundant flavonoids in both wild PI489777 and dormant RILs (Table 4 and Table 6). There were similarities between the concentrations in the cultivated vs. wild and non-dormant vs. dormant categories.

### 2.3. Phenylpropanoid Contents in Relation to Dormancy Status

The PLS-DA biplot (Figure 6) presents a closer look at the relationship between dormancy status and the amount of phenolic acids and flavonoids present. The first two PLS components explained 59.51% of the variability in phenolic acids and flavonoids. The coefficient of determination for the corresponding model was 0.67. Gallic acid was identified as the metabolite that distinguished most clearly between the non-dormant/cultivated parental genotype and dormant/wild parental genotype groups. Dormant CRIL2-27 deviated the most in this regard, with the amount of gallic acid most similar to non-dormant RILs, i.e., relatively high. There was greater overlap between the groups in the other metabolites with major discriminating effects, such as morin, kaempferol, and quercetin. In particular, these flavonoids are very abundant in dormant CRIL2-48 and CRIL2-21, and are rare in non-dormant CRIL2-65, CRIL2-81, CRIL2-80, CRIL2-45, and ICC4958, which is not in accordance with the majority of observations in the respective groups.

## 3. Discussion

Flavonoids and phenolic substances have been extensively analyzed in seeds and seed coats [10,41]. Contrasting pigmentations of seeds are found in many crops as a result of selection for less pigmented seeds during domestication [11,14,42,43,44]. Pigmentation has a protective role against biotic and abiotic stress; it adversely impacts palatability; and consequently, less pigmented seeds are desirable for cultivation and processing. However, with the increased interest in nutritional aspects, there has been increased interest in pigmentation, as many of these metabolites that contribute to increased pigmentation have health-promoting and antioxidant activities. While the content of secondary metabolites has been studied in seeds of legumes, these studies used only cultivated crops [45,46,47,48,49]. Comparisons of wild progenitors and the respective crops are much rarer [42]. The seed coats of chickpeas (and legumes in general) contain many phytochemicals, especially flavonoids and phenolic acids [8].

Connections between the expression of genes that determine proanthocyanidin and flavonoid biosynthetic pathways and seed dormancy, caused by a water-impermeable seed coat, were shown in *Arabidopsis* [40], *Medicago truncatula* Gaertn [50,51,52], and soybean [15]. According to Galussi et al. [53], several water-repelling substances that also join cell walls are responsible for physical dormancy. These substances include polyphenols, lignins, condensed tannins, and pectins, as well as some celluloses and hemicelluloses. Higher concentrations of polyphenolic substances were observed in colored seed coats of soybeans [54,55], lentils [56], peas [45,46,47,49], and chickpeas [48,49]. A relationship between the accumulation of proanthocyanidins, lignins, and the alteration of physical properties of the seed coat leading to cracking was found in soybean [55]. We found that the presence of epicatechin was positively related to the dormancy of seeds, as Zhou et al. [15] and Li et al. [38] demonstrated in soybeans. In addition to genetic factors, the contents of phenolics and flavonoids are influenced by environmental factors such as storage conditions and the access of seeds to oxygen. Moreover, Zhou et al. [15] directly tested the effect of particular phenolic compounds on the germination of soybeans and found a negative effect of epicatechin at higher concentrations.

Wild chickpea, *C. reticulatum* (PI489777), has medium to dark brown seeds and contained higher concentrations of 10 out of the 13 phenolic compounds that were detected than the genotype of the cultivated chickpea (ICC4958). Corresponding dormant RILs (considered as more pigmented) contained higher concentrations of phenolic compounds. In the case of flavonoids, only 4 out of the 13 detected substances (31%) had a higher concentration in the wild seeds of PI489777. This corresponded with non-dormant RILs (considered as less pigmented) having higher flavonoid contents than dormant RILs, similarly to cultivated ICC4958 having higher flavonoid contents than wild PI489777. However, the differences are not that distinct, and these relatively minor differences between wild and cultivated genotypes could be due to the cultivated parental genotype being of the *desi* type, which is more pigmented compared with the *kabuli* type, and thus contained higher contents of polyphenolic substances (13-fold more phenolic acids and 10–11-fold more flavonoids) [48]. Higher phenolic contents in wild and cultivated *desi* chickpeas in comparison to cultivated *kabuli* chickpeas were also found by Kaur et al. [57]. Moreover, there was also a difference in the rate of imbibing between *kabuli* and *desi* chickpea types. Non-pigmented *kabuli* seeds imbibed more quickly (within 4–8 h), whereas pigmented *desi* seeds imbibe more slowly [58] (within 24 h); this is partially related to differences in the thickness of their respective seed coats [59].

The most commonly detected phenolic acids in chickpeas were dihydroxybenzoic acid, *p*-coumaric acid, gallic acid, chlorogenic acid, ferulic acid, 4-hydroxybenzoic acid, and syringic acid. Among the most commonly detected flavonoids observed in chickpeas are pinocembrin, quercetin, catechin, luteolin, and myricetin [8,60,61,62]. Elessawy et al. [63] reported that chickpea and pea seed coats had the most similar compositions of polyphenolic substances among the legumes they compared. The most important phenolic acids and flavonoids in peas were protocatechin, vanillic acid, syringic acid, caffeic acid, ferulic acid, and *p*-coumaric acid [8,12,45]. Troszynska and Ciska [45] also observed differences in phenols between colored and white seed coats of peas, where protocatechuic, gentisic, and vanillic acids were dominant in colored peas, while hydroxycinnamic, ferulic, and coumaric acids were the most abundant in white-seeded peas. These findings correspond with our results, where vanillic acid was not detected in cultivated ICC4958 and several non-dormant, less pigmented seeds, while wild PI489777 and dormant RILs with more pigmented seeds had generally higher concentrations. A comparison of parental genotypes showed a higher concentration of coumaric acid in cultivated ICC4958, which corresponded to Troszynska and Ciska’s [45] findings of more coumaric acid in less pigmented genotypes. In contrast to Troszynska and Ciska [45], however, ferulic acid had higher levels in dormant RILs with more pigmented seed coats. Jha et al. [47] reported that gallic and caffeic acids are present only in seeds of a purple-flowered variety of pea and the corresponding RILs with pigmented seed coats, and are absent in varieties with white flowers and non-pigmented seeds and their corresponding RILs. They also mentioned the higher concentrations of epigallocatechin, vanillic, and 3,4-dihydroxybenzoic acids in the pigmented seed coats of pea. Our findings do not completely correspond with the results of Jha et al. [47]. However, these researchers worked only with domesticated, e.g., non-dormant genotypes, differing in pigmentation. Here, we detected *p*-coumaric acid, gallic acid, ferulic acid, 3-hydroxybenzoic and 4-hydroxybenzoic acid, syringic acid, catechin, quercetin, luteolin, and myricetin, corresponding to the results of Aguilera et al. [60], Fratianni et al. [61], and Magalhaes et al. [62]. According to Amarowicz and Pegg [64], legume seed coats primarily contain 4-hydroxybenzoic acid, protocatechin, gallic acid, vanillic acid, and syringic acid; however, vanillic acid was not detected in cultivated ICC4958 and nine RILs (regardless of the dormancy state). Gallic acid, which was mentioned above, was also not detected in several dormant RILs. In lentil genotypes, vanillic acid glucoside and gallic acid were identified as the most abundant phenolic acids in seed coats [65]; in terms of significance, both vanillic acid and gallic acid are ranked first in bean seed coats [8]. Lentil genotypes with colored seed coats are also abundant with flavonoids tricetin and luteolin [56]. On the other hand, syringic acid, which was previously identified as the primary substance found in legume seeds [64], was present in both parental chickpea genotypes and all derived RILs. In contrast to this finding, Elessawy et al. [63] did not identify syringic acid in the seed coats of the analyzed legume genotypes. This was further supported by Jha et al. [47], where none of the pea genotypes, regardless of seed coat pigmentation, contained syringic acid. In our study, we did not observe a significant difference in ferulic acid concentration between parental genotypes and RILs, which does not correspond with the finding of Jha et al. [47] where ferulic acid content was significantly higher in white flowered peas with non-pigmented seed coat. According to the study by Quintero-Soto et al. [66], the most abundant flavonoid in cultivated *desi* chickpea is catechin, which is assumed to be present mainly in colored chickpea seed coats. In our study, the catechin levels in cultivated ICC4958 were not very significant, but catechin was absent in wild PI489777 and several dormant RILs. This finding suggests that in the case of catechin, pigmentation probably does not play a crucial role; instead, catechin has been introduced to cultivated chickpeas by domestication. In contrast, a higher abundance of catechin was observed in wild soybeans [38]. Low-tannin species of lentil are also very low in levels of catechin [47,67].

All these metabolites are largely soluble, but there are also more complex polymerized insoluble phenolics in the seed coats of peas [11], faba beans [68], and lentils [69]. Although their direct involvement in seed-coat-imposed dormancy was not confirmed in peas [11], they still might contribute to this by providing substrates for lignin or suberin pathways for impregnating the seed coat.

Notable is the high content of salicylic acid and its glucoside in the seed coats of wild chickpea. Salicylic acid is known to be involved in plant responses to certain pathogens; it regulates diverse aspects of the plant’s responses to abiotic stresses through extensive interactions with other growth hormones [70]. Salicylic acid also plays a role in germination under stressful conditions, although its precise role and the underlying molecular mechanisms have not been fully elucidated.

We observed differences in the contents of selected phenolic acids and flavonoids between the seed coats of contrasting wild and cultivated chickpeas. We found that the differences were mainly in the presence of gallic acid, syringic acid, and sinapic acid for phenolic compounds, and quercetin, kaempferol, and naringenin for flavonoids. We noted particularly that the wild parental genotype and corresponding dormant RILs contained generally higher levels of phenolic acids and lower levels of flavonoids. The biosynthesis of phenolic acids is apparently more pronounced in dormant chickpea genotypes, and there is a possible relationship between the phenolic content and dormancy of the seeds. Metabolic and chemical compositions of the seed coats of chickpeas in relation to their germination, together with information about the anatomical structure, could be employed in further chickpea-breeding programs. This may prove especially useful in relation to dormancy-breaking mechanisms, which are essential for the successful establishment of crops. The presence of polyphenolic substances in the seed coats of chickpeas could also be part of plant breeding programs, due to their nutritional possibilities.

## 4. Materials and Methods

### 4.1. Seed Material

Seeds were obtained from Dr C. Coyne, USDA Pullman, USA, and originated from field-grown plants from the 2021 season on Central Ferry Farm, Pullman, WA, USA. Seed coats were dissected from dry seeds (approx. 7 seeds, depending on their size) of parental genotypes—cultivated non-dormant *C. arietinum* (ICC4958) and wild dormant *C. reticulatum* (PI489777). In addition, 16 non-dormant (CRIL2-5, CRIL2-6, CRIL2-7, CRIL2-15, CRIL2-23, CRIL2-43, CRIL2-45, CRIL2-47, CRIL2-50, CRIL2-51, CRIL2-65, CRIL2-80, CRIL2-81, CRIL2-89, CRIL2-110, and CRIL2-111) and 12 dormant (CRIL2-14, CRIL2-21, CRIL2-27, CRIL2-42, CRIL2-48, CRIL2-60, CRIL2-79, CRIL2-106, CRIL2-114, CRIL2-115, CRIL2-129, and CRIL2-131) recombinant inbred lines (RILs) were selected, as well as the CRIL2-25 line that exhibited intermediate behavior (i.e., neither dormant nor non-dormant). RILs were selected according to their dormancy level based on the study by Sedláková et al. [4].

### 4.2. Analysis of Phenolic Metabolites

Homogenized seed coats (≈20 mg) were mixed with 1 mL of solvent (acetone:water:acetic acid, 80:19:1) and treated for 10 min in an ultrasonic bath. After spinning in a centrifuge at 14,500× *g*, the supernatant was transferred into the new vial and kept at −20 °C until analyzed. Analysis of free phenolic acids and flavonoids was performed according to the protocol described in Ćavar Zeljković et al. [71,72]. Briefly, LC–MS/MS measurements were carried out using an Ultra Performance LCMS 8050 system (Shimadzu, Kyoto, Japan) with a triple quadrupole mass spectrometer equipped with an electrospray ionization (ESI) source operating in negative mode. The samples were injected (5 µL) into a reversed-phase column (Acquity UPLC BEH C18, 1.7 μm, 2.1 × 100 mm, Waters, Milford, MA, USA) with a corresponding pre-column (Acquity UPLC BEH C18 VanGuard pre-column, 1.7 μm, 2.1 mm × 5 mm). The mobile phase consisted of a mixture of 15 mM formic acid (pH 3, adjusted with NH_4_OH) (solvent A) and ACN (solvent B) at a flow rate of 0.4 mL/min. The linear gradient consisted of 10% B for 1 min, 10–13% B for 2 min, isocratic 13% B for 4 min, 13–25% B for 3 min, 25–70% B for 1.2 min, isocratic 75% B for 0.8 min, back to 10% B within 0.5 min, and equilibration for 3.5 min. The effluent was introduced into an electrospray source (interface temperature of 300 °C, heat block temperature of 400 °C, and capillary voltage of 3.0 kV). To achieve high specificity in addition to high sensitivity, the analysis was performed in multiple reaction monitoring (MRM) mode. All standards and reagents were of the highest available purity and purchased from Sigma Aldrich Company (Prague, Czech Republic).

In total, 50 compounds were analyzed, representing all classes of phenylpropanoids, i.e., 9 hydroxybenzoates (gallic acid, 3,4-dihydroxybenzoic acid, salicylic acid-2-O-β-d-glucoside, 2,3-dihydroxybenzoic acid, 4-hydroxybenzoic acid, 3-hydroxybenzoic acid, vanillic acid, syringic acid, and salicylic acid), 14 hydroxycinnamates (5-hydroxyferulic acid, chlorogenic acid, caffeic acid, *p*-coumaric acid, sinapic acid, ferulic acid, rosmarinic acid, *trans*-cinnamic acid, *p*-methyl coumarate, coniferyl alcohol, sinapyl alcohol, coniferaldehyde, sinapaldehyde, and abietin), 1 chalcone (phloretin), 3 flavanols (gallocatechin, epigallocatechin, and catechin), 8 flavonols (myricetin, quercetin, morin, kaempferol, galangin, myricitrin, rutin, and quercitrin), 10 flavones (luteolin, apigenin, chrysoeriol, chrysin, cannflavin A, cannflavin B, orientin, isoorientin, vitexin, and isovitexin), and 5 flavanones (naringenin, eriodictyol, pinocembrin, naringin, and hesperidin). All measurements were performed in triplicate.

### 4.3. Statistical Analyses

Statistical analyses were performed in RStudio (R Software ver. 4.1.0) using the packages *corrplot*, *ggplot2*, and *pls*. Values for phenolic acids and flavonoids that were not detected were replaced by two-thirds of the minimal detected value in the respective variable in order to carry out the subsequent statistical analysis. The relative differences (in log-scale) in the content of phenolic acids and flavonoids between two parental genotypes were computed and shown with barplots. In addition to the differences in the mean values of the three replicates, the largest and smallest differences in the individual replicates were displayed; thus, the significant difference could be calculated (i.e., points below zero (and points above zero) indicate that the corresponding Wilcoxon test yields a *p*-value of 0.1, the smallest achievable *p*-value with this test when comparing two groups with a sample size of 3). The Wilcoxon test was applied to determine the differences between the non-dormant and dormant RILs groups. Boxplots for the phenolic acids and flavonoids showing significant differences (at a statistical significance level α = 0.05) between the groups were derived. The data were log-transformed and a PLS-DA biplot was constructed (taking dormancy as the response).

## Figures and Tables

**Figure 1 plants-12-02687-f001:**
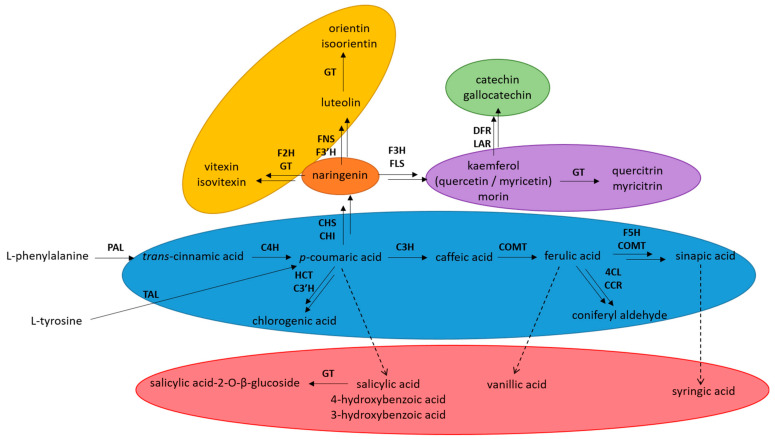
Simplified phenylpropanoid pathway. PAL—phenylalanine ammonia-lyase, TAL—tyrosine ammonia-lyase, C4H—cinnamate 4-hydroxylase, C3H—4-coumarate 3-hydroxylase, COMT—cinnamyl (caffeate) *O*-methyltransferase, F5H—ferulate (coniferyl aldehyde/alcohol) 5-hydroxylase, HCT—*p*-hydroxycinnamoyl-CoA: quinate/shikimate *p*-hydroxycinnamoyl transferase, C′3H—*p*-coumaroyl-shikimate/quinate 3-hydroxylase, 4CL—4-coumarate-CoA: ligase, CCR—cinnamoyl-CoA reductase, FNS—flavone synthase II, F3′H—flavonoid 3′-monooxigenase, CHS—chalcone synthase, CHI—chalcone isomerase, F3H—flavonoid-3-hydroxylase, F2H—flavonoid-2-hydroxylase, GT-glycosyltransferase, DFR—dihydroflavanol 4-reductase, LAR—leucocyanidin reductase. The solid arrows indicate pathways described in the literature; the dashed arrows indicate transformations that are still unclear.

**Figure 2 plants-12-02687-f002:**
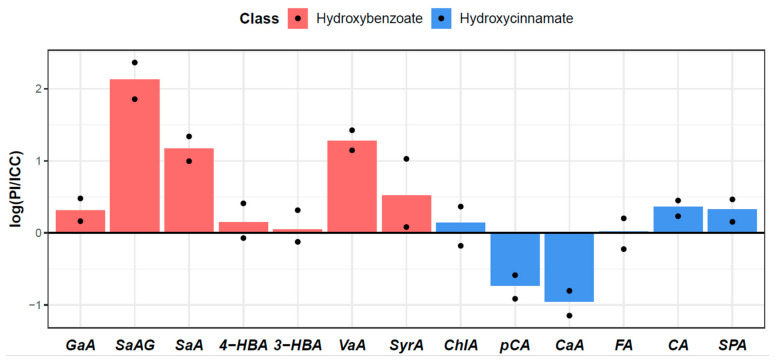
Barplot of relative differences (in log-scale) in phenolic acids between the parental genotypes of cultivated ICC4958 and wild chickpea PI489777. The points represent the highest and the lowest difference with regard to the individual replicates. Thus, both points under zero and points over zero indicate significant differences. Hydroxybenzoate: GaA—gallic acid; SaAG—salicylic acid-2-O-β-d-glucoside; SaA—salicylic acid; 4-HBA—4-hydroxybenzoic acid; 3-HBA—3-hydroxybenzoic acid; VaA—vanillic acid; SyrA—syringic acid; hydroxycinnamate: ChlA—chlorogenic acid; pCA—*p*-coumaric acid; CaA—caffeic acid; FA—ferulic acid; CA—coniferaldehyde; SPA—sinapic acid.

**Figure 3 plants-12-02687-f003:**
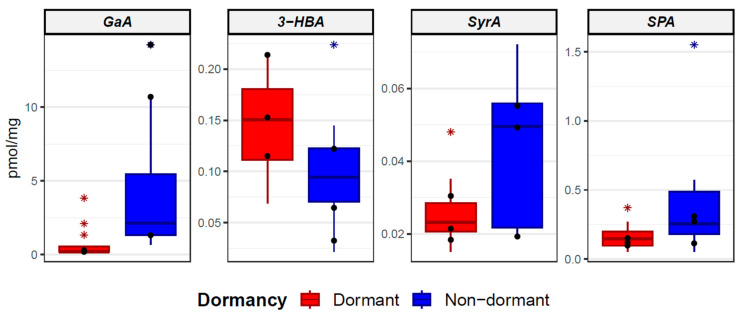
Boxplots of contrasting phenolic acids compared between non-dormant and dormant RILs. GaA—gallic acid; 3-HBA—3-hydroxybenzoic acid; SyrA—syringic acid; SPA—sinapic acid. Black dots represent 3 selected dormant RILs (CRIL2-106, CRIL2-60, and CRIL2-131) and 3 selected non-dormant RILs (CRIL2-5, CRIL2-6, and CRIL2-50).

**Figure 4 plants-12-02687-f004:**
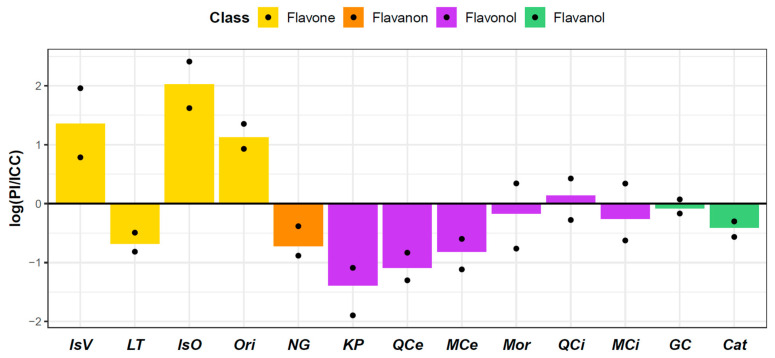
Barplots of relative differences (in log scale) in flavonoids between the parental genotypes of cultivated ICC4958 and wild PI489777. The points represent the highest and the lowest difference with regard to the individual replicates. Thus, both points under zero and points over zero indicate significant differences. Flavone: IsV—isovitexin; LT—luteolin; IsO—isoorientin; Ori—orientin; Flavanon: NG—naringenin; Flavonol: KP—kaempferol; QCe—quercetin; MCe—myricetin; Mor—morin; QCi—quercitrin; MCi—myricitrin; Flavanol: GC—gallocatechin; Cat—catechin.

**Figure 5 plants-12-02687-f005:**
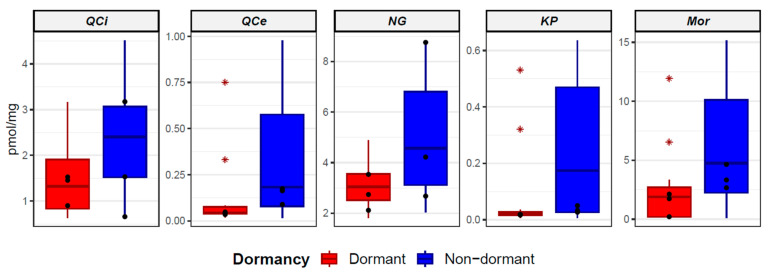
Boxplots of contrasting flavonoids compared between non-dormant and dormant RILs. QCi—quercitrin; QCe—quercetin; NG—naringenin; KP—kaempferol; Mor—morin. Black dots represent 3 selected dormant RILs (CRIL2106, CRIL2-60, and CRIL2-131) and 3 selected non-dormant RILs (CRIL2-5, CRIL2-6, and CRIL2-50).

**Figure 6 plants-12-02687-f006:**
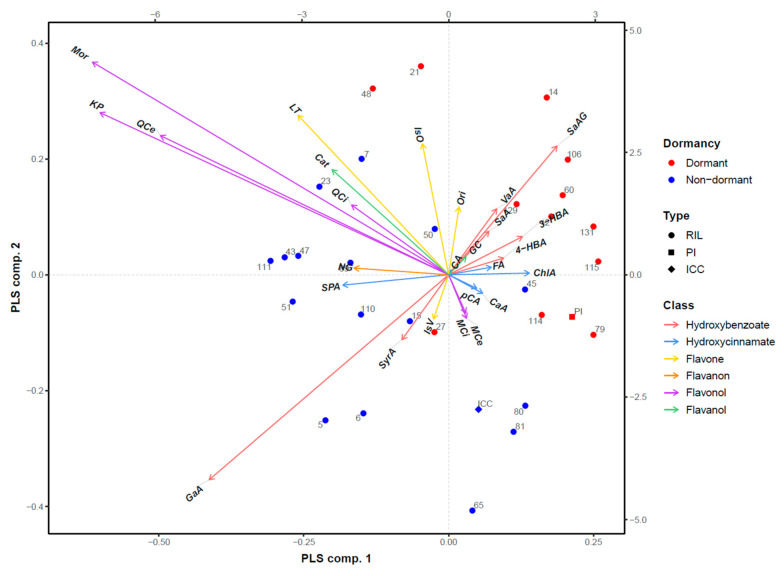
PLS-DA biplot presenting the relationship between the dormancy status and the levels of phenolic acids and flavonoids. RIL—recombinant inbred line; PI—wild PI489777; ICC—cultivated ICC4958; Hydroxybenzoate (phenolics): GaA—gallic acid; SaAG—salicylic acid-2-O-β-d-glucoside; SaA—salicylic acid; 4-HBA—4-hydroxybenzoic acid; 3-HBA—3-hydroxybenzoic acid; VaA—vanillic acid; SyrA—syringic acid; Hydroxycinnamate (phenolics): ChlA—chlorogenic acid; pCA—*p*-coumaric acid; CaA—caffeic acid; FA—ferulic acid; CA—coniferaldehyde; SPA—sinapic acid. Flavone (flavonoids): IsV—isovitexin; LT—luteolin; IsO—isoorientin; Ori—orientin; Flavanon (flavonoids): NG—naringenin; Flavonol (flavonoids): KP—kaempferol; QCe—quercetin; MCe—myricetin; Mor—morin; QCi—quercitrin; MCi—myricitrin; Flavanol (flavonoids): GC—gallocatechin; Cat—catechin.

**Table 1 plants-12-02687-t001:** Phenolic compounds (pmol/mg DW) present in the seed coats of parental genotypes (cultivated non-dormant ICC4958, wild dormant PI489777).

		ICC4958		PI489777		
Class	Compound	Mean	SD	Mean	SD	
Hydroxybenzoate	Gallic acid	1.715	0.157	2.336	0.156	*
	Salicylic acid-2-O-β-d-glucoside	0.508	0.033	4.258	0.809	*
	4-Hydroxybenzoic acid	8.303	0.865	9.645	1.350	
	Vanillic acid	–	–	0.929	0.136	*
	3-Hydroxybenzoic acid	0.088	0.015	0.092	0.005	
	Syringic acid	0.025	0.006	0.043	0.011	*
	Salicylic acid	0.580	0.035	1.870	0.226	*
Hydroxycinnamate	Chlorogenic acid	0.438	0.033	0.504	0.100	
	Caffeic acid	0.054	0.002	0.021	0.003	*
	*p*-Coumaric acid	0.389	0.002	0.187	0.029	*
	Ferulic acid	0.678	0.026	0.692	0.119	
	Sinapic acid	0.114	0.004	0.159	0.019	*
	Coniferaldehyde	0.031	0.001	0.045	0.004	*

* Relative differences between cultivated and wild parental genotypes are significant.

**Table 2 plants-12-02687-t002:** Average values of phenolic compounds (pmol/mg DW) detected in the seed coats of non-dormant chickpea RILs.

Class	Compound	Mean	SD	Min	Max	
Hydroxybenzoate	Gallic acid	3.814	3.814	0.673	14.240	*
	Salicylic acid-2-O-β-d-glucoside	2.720	2.133	0.110	8.148	
	4-Hydroxybenzoic acid	7.150	2.797	2.163	11.239	
	Vanillic acid	0.6927	0.230	0.476	1.225	
	3-Hydroxybenzoic acid	0.103	0.046	0.032	0.224	*
	Syringic acid	0.043	0.018	0.019	0.072	*
	Salicylic acid	1.283	0.913	0.244	2.960	
Hydroxycinnamate	Chlorogenic acid	0.296	0.147	0.128	0.633	
	Caffeic acid	0.036	0.014	0.014	0.066	
	*p*-Coumaric acid	0.221	0.073	0.106	0.427	
	Ferulic acid	0.659	0.346	0.217	1.499	
	Sinapic acid	0.369	0.357	0.054	1.551	*
	Coniferaldehyde	0.038	0.021	0.015	0.080	

* Statistically significant difference (*p* < 0.05) between non-dormant and dormant RILs.

**Table 3 plants-12-02687-t003:** Average values of phenolic compounds (pmol/mg DW) detected in the seed coats of dormant chickpea RILs.

Class	Compound	Mean	SD	Min	Max	
Hydroxybenzoate	Gallic acid	1.893	1.492	0.297	3.837	*
	Salicylic acid-2-O-β-d-glucoside	3.646	1.357	1.606	5.787	
	4-Hydroxybenzoic acid	8.289	3.469	4.518	12.974	
	Vanillic acid	0.948	0.509	0.389	1.904	
	3-Hydroxybenzoic acid	0.146	0.048	0.069	0.214	*
	Syringic acid	0.026	0.009	0.015	0.048	*
	Salicylic acid	1.145	0.492	0.396	2.291	
Hydroxycinnamate	Chlorogenic acid	0.418	0.191	0.103	0.806	
	Caffeic acid	0.043	0.029	0.012	0.095	
	Ferulic acid	0.267	0.094	0.140	0.446	
	*p*-Coumaric acid	0.957	0.582	0.303	1.991	
	Sinapic acid	0.162	0.091	0.053	0.370	*
	Coniferaldehyde	0.052	0.025	0.014	0.095	

* Statistically significant difference (*p* < 0.05) between non-dormant and dormant RILs.

**Table 4 plants-12-02687-t004:** Flavonoid levels (pmol/mg DW) detected in the seed coats of parental genotypes (cultivated non-dormant ICC4958, wild dormant PI489777).

		ICC4958		PI489777		
Class	Compound	Mean	SD	Mean	SD	
Flavanol	Gallocatechin	11.196	1.890	10.309	0.224	
	Catechin	0.273	0.041	–	–	*
Flavonol	Myricetin	17.668	0.895	7.806	1.619	*
	Quercetin	0.064	0.001	0.021	0.005	*
	Kaempferol	0.040	0.006	0.010	0.003	*
	Morin	0.136	0.025	0.115	0.040	
	Myricitrin	0.892	0.294	0.690	0.097	
	Quercitrin	0.720	0.114	0.827	0.169	
Flavone	Luteolin	0.349	0.033	0.177	0.013	*
	Isoorientin	0.007	0.001	0.050	0.012	*
	Orientin	0.028	0.003	0.085	0.009	*
	Isovitexin	0.005	0.001	0.020	0.006	*
Flavonone	Naringenin	4.740	0.984	2.310	0.130	*

* Relative differences between cultivated and wild parental genotypes are significant.

**Table 5 plants-12-02687-t005:** Average values of flavonoids (pmol/mg DW) detected in the seed coats of non-dormant chickpea RILs.

Class	Compound	Mean	SD	Min	Max	
Flavanol	Gallocatechin	10.043	4.078	5.720	19.461	
	Catechin	0.739	0.171	0.421	0.917	
Flavonol	Myricetin	8.319	4.161	2.542	20.010	
	Quercetin	0.321	0.302	0.016	0.977	*
	Kaempferol	0.248	0.243	0.006	0.635	*
	Morin	6.205	4.974	0.109	15.149	*
	Myricitrin	1.290	0.547	0.581	2.541	
	Quercitrin	2.416	1.186	0.632	4.510	*
Flavone	Luteolin	1.220	0.989	0.126	3.275	
	Isoorientin	0.015	0.014	0.003	0.056	
	Orientin	0.036	0.021	0.017	0.096	
	Isovitexin	0.007	0.010	0.001	0.041	
Flavonone	Naringenin	5.014	2.265	2.042	8.826	*

* Statistically significant difference (*p* < 0.05) between non-dormant and dormant RILs.

**Table 6 plants-12-02687-t006:** Average values of flavonoids (pmol/mg DW) detected in the seed coats of dormant chickpea RILs.

Class	Compound	Mean	SD	Min	Max	
Flavanol	Gallocatechin	11.833	3.851	5.621	17.576	
	Catechin	0.850	0.495	0.452	1.587	
Flavonol	Myricetin	8.407	3.327	2.251	15.143	
	Quercetin	0.129	0.213	0.020	0.751	*
	Kaempferol	0.088	0.164	0.007	0.531	*
	Morin	2.709	3.425	0.114	11.932	*
	Myricitrin	1.205	0.767	0.393	3.049	
	Quercitrin	1.497	0.867	0.635	3.163	*
Flavone	Luteolin	1.125	1.567	0.337	5.925	
	Isoorientin	0.022	0.021	0.003	0.073	
	Orientin	0.049	0.037	0.017	0.151	
	Isovitexin	0.004	0.002	0.002	0.008	
Flavonone	Naringenin	3.038	0.869	1.814	4.880	*

* Statistically significant difference (*p* < 0.05) between non-dormant and dormant RILs.

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
