# Peer review of "Phenylpropanoid Content of Chickpea Seed Coats in Relation to Seed Dormancy"

_plants, 2023, doi:10.3390/plants12142687_

Round 1

Reviewer 1 Report

The authors should reformulate some sentences. E.g. row 137 the authors state that the most abundant compound for both genotypes is 4 hydroxybenzoic acid and in row 147 they state that"on contrary" the most abundant compound was 4 hydroxybenzoic acid. 

Row 613 correct the formula for ammonium hydroxide.

Some sentences should be reformulated, e.g. row 128-130. Sentences should start with the subject.

Reviewer 2 Report

Minor editing of English language required

Reviewer 3 Report

The manuscript is valuable and provides additional information regarding both basic and practical knowledge.

Lines 115-118: maybe it would be worth presenting the assumptions of the research in the form of hypotheses

Lines 118-120 ‘The content  of selected phenolic acids and flavonoids in the chickpea seed coats was analyzed via ultra-performance liquid chromatography coupled with tandem mass spectrometry  (LC-MS/MS).’ I would put this information in methods.

Line 123: ‘The chickpea dormancy status was determined recently published paper by  Sedláková et al. [4]’. although the Authors refer to the publication, I would give brief information.

Lines 570-584: This part of the manuscript is a summary of the research. The information contained therein is interesting and valuable, and it is a pity that it is not a form of verifying the hypothesis.
